# Consolidator: Mergeable Adapter with Grouped Connections for Visual Adaptation

**Tianxiang Hao**[1,3]**, Hui Chen**[2,3†]**, Yuchen Guo**[3]**, Guiguang Ding**[1,3†]
[1]School of Software, Tsinghua University  [2]Department of Automation, Tsinghua University
[3]Beijing National Research Center for Information Science and Technology (BNRist)
{beyondhtx,jichenhui2012,yuchen.w.guo}@gmail.com, dinggg@tsinghua.edu.cn

## Abstract

Recently, transformers have shown strong ability as visual feature extractors, surpassing traditional convolution-based models in various scenarios. However, the success of vision transformers largely owes to their capacity to accommodate numerous parameters. As a result, new challenges for adapting a well-trained transformer to downstream tasks arise. On the one hand, classic fine-tuning tunes all parameters in a huge model for every downstream task and thus easily falls into an overfitting situation, leading to inferior performance. On the other hand, on resource-limited devices, fine-tuning stores a full copy of all parameters and thus is usually impracticable for the shortage of storage space. However, few works have focused on how to efficiently and effectively transfer knowledge in a vision transformer. Existing methods did not dive into the properties of visual features, leading to inferior performance. Moreover, some of them bring heavy inference cost though benefiting storage. To tackle these problems, we propose consolidator to achieve efficient transfer learning for large vision models. Our consolidator modifies the pre-trained model with the addition of a small set of tunable parameters to temporarily store the task-specific knowledge while freezing the backbone model during adaptation. Motivated by the success of group-wise convolution, we adopt grouped connections across the features extracted by fully connected layers to construct tunable parts in a consolidator. To further enhance the model's capacity to transfer knowledge under a constrained storage budget and keep inference efficient, we consolidate the parameters in two stages: 1. between adaptation and storage, and 2. between loading and inference. On a series of downstream visual tasks, our consolidator can reach up to 7.56 better accuracy than full fine-tuning with merely 0.35% parameters, and outperform state-of-the-art parameter-efficient tuning methods by a clear margin. Code is available at github.

## 1 Introduction

Recently, transformer architectures originated from natural language processing (NLP) (Vaswani et al., 2017) demonstrate considerable capacity in computer vision (Dosovitskiy et al., 2020; Touvron et al., 2021; Liu et al., 2021b). Vision transformers, along with traditional convolutional neural networks (CNNs) (Krizhevsky et al., 2012; He et al., 2016; Simonyan & Zisserman, 2014), are widely used as feature extractors to generate strong and general visual representations via deriving knowledge from massive images. Thanks to the abundant information in such representations, we can adapt the pre-trained models to downstream tasks by a simple fine-tuning strategy.

However, fine-tuning is not a good solution for adaptation. As is well known, the scale of vision models grows faster and faster in recent years. On the one hand, fine-tuning which tunes all parameters in such a huge model easily falls into an overfitting situation, leading to inferior performance. On the other hand, fine-tuning inflicts heavy storage burdens. Since fine-tuning intensively tunes all parameters, it maintains a full copy of the model's parameters for each task. Therefore, fine-tuning can cause a huge storage burden when there are many tasks to be adapted, resulting in impracticality in real-world scenarios, especially in resource-constrained situations, *e.g.*, embedded systems.

---

[†]Corresponding authors.

Efforts have been made to improve the performance as well as reduce the storage overhead of fine-tuning. For example, adapter (Houlsby et al., 2019; Karimi Mahabadi et al., 2021), prompt tuning (Li & Liang, 2021; Lester et al., 2021; Zhou et al., 2021) and LoRA (Hu et al., 2021) inject tunable parameters and freeze the backbone during adaptation. In the vision field, VPT (Jia et al., 2022) directly leverage learnable prompts, AdaptFormer (Chen et al., 2022) adopts parallel adapter, NOAH (Zhang et al., 2022) searches for the optimal combinations of the three representative modules, *i.e.*, adapter, LoRA, and VPT, and SSF (Lian et al., 2022b) use additional scaling and shifting parameters for adaptation. Despite their acceptable performance, existing methods suffer from two common conflicts: 1. trade-off between the inference efficiency and the adaptation performance, and 2. trade-off between the adaptation performance and the number of stored parameters. Previous works (Houlsby et al., 2019) show that introducing more tunable parameters can achieve more fruitful results. However, extra parameters can bring significantly larger computation and storage cost, resulting in low inference efficiency and more storage space. Therefore, one essential question is raised: *can we design a module that can share the same inference cost as an ordinary model while enjoying superior capacity against existing methods*?

In this paper, we propose a generic module, dubbed consolidator, to tackle the aforementioned issues. The proposed consolidator is designed as a mergeable adapter that accompanies the fully connected (FC) layer in the vision models. Specifically, to enrich the model capacity under a limited parameter budget, we take inspiration from the success of group-wise convolution (Howard et al., 2017; Ma et al., 2018; Liu et al., 2022) and build our consolidator as grouped connected (GC) layers. To enhance the flexibility, we further reorder channels for each group connection, followed by a droppath regularizer. Benefiting from the inference-time linearity of GC, channel reorder, and droppath operations, the proposed consolidator can be perfectly consolidated into the original FC layer of a vision model, leading to no extra inference cost.

Our consolidator can be easily expanded as a multi-branch topology without breaking the linearity. Practically, we can simultaneously equip several GC layers with channel reordering for communications between different groups of feature channels. After adaptation, we can first consolidate the multi-branch GC layers into one single sparse parameter matrix and store the sparse matrix for each task. Such property can enhance the model's transferability and achieve a considerable storage reduction when the number of tasks scales up. During inference, such a sparse parameter matrix can be merged into the backbone model as well, resulting in no inference cost. Thanks to the twice consolidation, the proposed consolidator can greatly promote efficient and effective visual adaptation.

To verify the superiority of consolidator, we conduct extensive experiments and analysis on a series of downstream recognition tasks. Experimental results show that our consolidator can surpass full fine-tuning by 7.56 top-1 accuracy with merely 0.35% parameters per task. Compared to state-of-the-art methods, such as NOAH, AdaptFormer and SSF our method can consistently reach better performance while enjoying no inference cost. On other fundamental visual tasks, *i.e.*, object detection and semantic segmentation, our consolidator shows great power as well.

Overall, we summarize our contributions as follows. (i) We propose a basic module, dubbed consolidator, for effective and efficient visual transfer learning. To enhance the transferability under limited tunable parameters, our consolidator is designed as a mergeable grouped connected (GC) layer with a channel reorder layer and a droppath regularizer. We extend the single branch to a multi-branch topology for better flexibility and transferability. (ii) We design a two-stage consolidation scheme by merging corresponding parameters in the training-storage phase and loading-inference phase. In this way, we can maximally dig the adaptation capacity of the model under a constrained storage budget, with no extra inference cost. (iii) We conduct extensive experiments and analysis on various downstream tasks. Results show that the proposed consolidator method can consistently outperform state-of-the-art methods with fewer stored parameters but superior performance.

## 2 RELATED WORKS

**Parameter-efficient transfer learning.** In the language field, works (Houlsby et al., 2019; Pfeiffer et al., 2021; Li & Liang, 2021; Lester et al., 2021; Zaken et al., 2021; Hu et al., 2021; Karimi Mahabadi et al., 2021; Liu et al., 2021a; Ding et al., 2022a) have been done to efficiently transfer the knowledge of pre-trained transformers to downstream language tasks. In the field of visual adaptation, several explorations have also been made to adapt vision transformers efficiently. Jia et al.

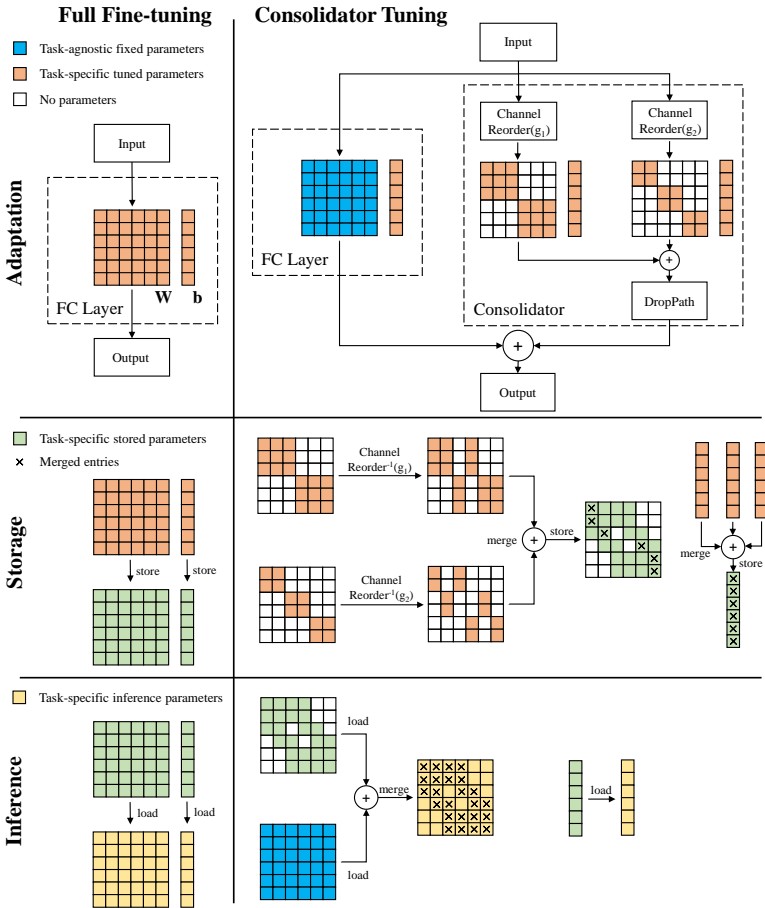

Figure 1: Consolidator tuning versus full fine-tuning. Consolidator adds tunable multi-branch grouped connected layers to the original fully connected layers. The tunable parameters are merged via addition into one single sparse matrix before storage to reduce the needed storage space. Between loading and inference, the parameters in the sparse matrix will be merged back into the original fully connected layer. Consolidator greatly enlarges the model's adaptation capacity under a constrained storage budget with no extra inference cost. **Best viewed in color.**

(2022) and Bahng et al. (2022) directly apply prompt-tuning. Jie & Deng (2022) integrates additional tunable convolution layers. NOAH (Zhang et al., 2022) first trains a large supernet with three modules, VPT, LoRA, and adapter, and then searches for the optimal configurations of each module for every transformer block using evolution algorithm (Chen et al., 2021). AdaptFormer (Chen et al., 2022) adds parallel adapters instead of serial ones. SSF (Lian et al., 2022b) tunes additional scaling and shifting parameters for adaptation. It is also shown that classic methods such as LoRA (Hu et al., 2021) and adapter (Houlsby et al., 2019) can lead to good performance for vision transformers. However, existing methods suffer from the two trade-offs as we discussed in Section 1, resulting in difficulties in fully digging the adaptation capacity of vision models efficiently. To solve the problems, we present a mergeable adapter, named consolidator, and introduce a two-stage consolidation design to perfectly balance the trade-offs, leading to efficient and effective visual adaptation.

**Inference-efficient structures.** Many works (Ding et al., 2019; Guo et al., 2020; Ding et al., 2021b;c;a; 2022b) strive to design a generic convolution architecture to realize superior capacity while enjoying no inference cost. For example, RepVGG (Ding et al., 2021c) integrates an extra 1×1 convolution to strengthen the main 3×3 convolution. However, existing methods are typically designed for CNNs. As for the popular vision transformer architectures, rare works investigate how to effectively strengthen their capacity while introducing no extra inference cost. LoRA (Hu et al., 2021) and SSF (Lian et al., 2022b) offer possible solutions, but they do not explore the consolidation process between training and storage, leading to inferior performance under a given storage budget. In this paper, we adopt parallel GC layers to replace the functionality of the original FC

layers in vision models, which shows strong abilities for visual adaptation. Furthermore, we expand the existing one-stage training-inference consolidation to a two-stage process: 1. training-storage consolidation, and 2. loading-inference consolidation. Such a two-stage design can maximally dig the adaptation capacity of the pre-trained model under a constrained storage budget, with no extra inference cost. Extensive experiments show that our consolidator can outperform state-of-the-art methods in both the number of tunable parameters and the adaptation performance.

## 3 METHODOLOGY

### 3.1 PRELIMINARIES

In this paper, we mainly focus on the adaptation for vision transformers (Dosovitskiy et al., 2020; Liu et al., 2021b). A typical vision transformer (Dosovitskiy et al., 2020) consists of $L$ serial blocks. In each encoder, there are a multi-head self-attention module (MHSA) and a multi-layer perceptron (MLP). Formally, a batch of input images $\mathbf{x}_{input} \in \mathbb{R}^{B \times 3 \times H \times W}$ will be first reshaped into a sequence of flattened 2D patches $\mathbf{x}_p \in \mathbb{R}^{B \times N \times (P^2 \cdot C)}$, where $C$ is the number of channels and $(P, P)$ is the resolution of each patch, and $N = NW/P^2$ is the number of patches. Then the patches are mapped to $D$ channel dimensions with a linear projection. Next, a classification token is appended and we can get $\mathbf{x}_1 \in \mathbb{R}^{B \times (N+1) \times D}$. Here we use $\mathbf{x}_l \in \mathbb{R}^{B \times (N+1) \times D}$ to denote the input of $l$-th ($1 \leq l \leq L$) block. Its output $\mathbf{x}_{l+1} = \mathbf{x}'_l + \text{MLP}(\text{LayerNorm}(\mathbf{x}'_l))$ where $\mathbf{x}'_l = \mathbf{x}_l + \text{MHSA}(\text{LayerNorm}(\mathbf{x}_l))$. For MHSA, the input features are first processed by three FC layers to generate matrices $Q, K, V$, and the output is calculated by $\text{Softmax}(\frac{QK^T}{\sqrt{d}})V$ and then projected by another FC layer. Therefore, the parametric components of MHSA are four FC layers. The parametric components of MLP are two FC layers as well. Therefore, we formulate our consolidator for the FC layers (see Fig. 1), covering all the parametric components in each MHSA and MLP. We will show that such a design can realize both efficiency and effectiveness in Section 4. Notably, our method is also applicable for MLP (Lian et al., 2022a) and CNN (Liu et al., 2022) and can reach good results as in Tab. 4.

### 3.2 CONSOLIDATOR

For efficient transfer learning, we merely tune and store the parameters in consolidators while freezing other parameters in the pre-trained model. In this subsection, we will introduce our design of consolidator, an efficient and effective module for adapting vision transformers, in detail.

**Grouped connections.** Inspired by the success of group convolution in extracting visual features, we hence assume that the cross-channel information exchange is redundant in visual adaptation, and aim to design consolidator by reducing the cross-channel connections between sequential features to minimize the number of stored parameters for downstream tasks while keeping maximum capacity. Therefore, for each FC layer, we add a concurrent module consisting of a grouped connected layer. Formally, for an input $\mathbf{x} \in \mathbb{R}^D$, the output $\mathbf{x}' \in \mathbb{R}^E$ of a GC layer with group $g$, weight $\mathbf{W} \in \mathbb{R}^{g \times \frac{E}{g} \times \frac{D}{g}}$ and bias $\mathbf{b} \in \mathbb{R}^E$ is formulated by $\mathbf{x}' = \text{GC}(\mathbf{x}) = \sum_{j=1}^{g} \text{Pad}(\mathbf{W}_j \mathbf{x}_{\frac{(j-1)D}{g} : \frac{jD}{g}}, j) + \mathbf{b}$.

Here $\text{Pad}(\mathbf{z}, j)$ prepends $\frac{(j-1)D}{g}$ zeros and appends $\frac{(g-j)D}{g}$ zeros for $\mathbf{z} \in \mathbb{R}^{\frac{E}{g}}$ according to another input $j$. In this way, the output channels in the $j$-th group only interact with the input channels in the $j$-th group, and thus we reduce the cross-channel connections as expected.

To flexibly reach different ratios of stored parameters, we adopt a multi-branch topology in our consolidator. There is a GC layer with weight $\mathbf{W}^{(i)} \in \mathbb{R}^{g^{(i)} \times \frac{E}{g^{(i)}} \times \frac{D}{g^{(i)}}}$ and bias $\mathbf{b}_i \in \mathbb{R}^E$ for $i$-th branch with group $g^{(i)}$. During adaptation, consolidator and the original FC layer take the same input and their outputs are summed up to produce the new output $\mathbf{y}$. Formally, for each FC layer with weight $\mathbf{W} \in \mathbb{R}^{E \times D}$ and bias $\mathbf{b} \in \mathbb{R}^E$, the output of the whole layer modified by $m$ GC branches is

$$\mathbf{y} = \mathbf{W}\mathbf{x} + \mathbf{b} + \sum_{i=1}^{m} \left( \sum_{j=1}^{g^{(i)}} \text{Pad}(\mathbf{W}_j^{(i)} \mathbf{x}_{\frac{(j-1)D}{g^{(i)}} : \frac{jD}{g^{(i)}}}, j) + \mathbf{b}^{(i)} \right).$$

**Channel reorder.** To flexibly tune the total number of parameters and enrich the exchange of information flow, we prepend a "ChannelReorder" operation to every branch in our consolidator by

manually adjusting the permutation of input features along the channel dimension. In general, we adopt shuffle operation (Zhang et al., 2018; Ma et al., 2018) to accomplish such a purpose.

Formally, given input $\mathbf{x} \in \mathbb{R}^{* \times D}$ where "$*$" means any number of dimensions including none, we shuffle it into $g$ groups and perform recombination across the last dimension. Formally, we first reshape $\mathbf{x}$ into $\mathbf{x}' \in \mathbb{R}^{* \times g \times \frac{D}{g}}$, and then transpose the last two dimension and get $\mathbf{x}'' \in \mathbb{R}^{* \times \frac{D}{g} \times g}$, and then reshape $\mathbf{x}''$ into $\mathbf{x}''' \in \mathbb{R}^{* \times D}$, which is the final output. A pythonic style formulation is $\text{ChannelReorder}(g, \mathbf{x}) = (\mathbf{x}.\text{reshape}(*, g, \frac{D}{g})).\text{transpose}(-2, -1).\text{reshape}(*, D)$

We set shuffle groups $g = g_i$ in the $i$-th branch, where $g_i$ is the group of the corresponding GC layer. In this way, there are few overlaps between the weight matrices of distinct branches, and the model capacity is greatly expanded each time a new branch is contained.

**Stochastic depth of pre-trained weight.** To further enlarge the model's adaptation capacity, we append a droppath (Huang et al., 2016) layer to each branch. For small downstream datasets, dropping the consolidator path with a higher ratio $p$ can help reduce overfitting and catastrophic forgetting, which may be beneficial to the performance. Empirically shown in Section 4.8, droppath is more effective than standard dropout (Srivastava et al., 2014) in the current situation probably for the following reasons. The parameters of the whole layer degrade into the pre-trained weight parameters with probability $p$. The frozen pre-trained parameters contain domain-generic knowledge, which may help the adaptation. Overall, a model modified by consolidator will have a stochastic depth of pre-trained state of parameters during each forward pass, and different consolidators will be activated for different training samples.

**Two-stage consolidation.** Now we have constructed all the elements of a consolidator. Formally, the output of the whole layer after being modified by consolidator is $\mathbf{y} = \mathbf{Wx} + \mathbf{b} + \text{Droppath}(p, \sum_{i=1}^{m} (\sum_{j=1}^{g^{(i)}} \text{Pad}(\mathbf{W}_j^{(i)} \text{ChannelReorder}(g^{(i)}, \mathbf{x})_{\frac{(j-1)D}{g^{(i)}} : \frac{jD}{g^{(i)}}}, j) + \mathbf{b}^{(i)}))$.

Since all the operations in a consolidator are inference-time linear, we can easily consolidate the domain-specific knowledge into the domain-agnostic knowledge in the pre-trained backbone model, in both the training-storage phase and the loading-inference phase.

1. Training-storage consolidation. All we need to store in a consolidator are $\mathbf{W}^{(i)}$ and $\mathbf{b}^{(i)}$. However, there are some parameters corresponding to the same entry, and thus they can be merged into a single one. As shown in Fig. 1, we also tune the bias of the original FC layer in addition to the parameters in consolidator. It is easy to find that the duplicate biases in all branches and the original bias can be merged into a single one. And there are some duplicate entries in the weight matrix as well, so we can merge all weight matrices into one single sparse matrix. Consolidating such duplicate entries can largely benefit storage. Formally, we use $\widetilde{\mathbf{W}}$ and $\widetilde{\mathbf{b}}$ to denote the matrix that we need to store on the disk. Since channel reorder is a linear operation, we can apply a reverse operation to $\mathbf{W}^{(i)}$ to simulate the effect of the reorder applied to $\mathbf{x}$. And we have $\widetilde{\mathbf{W}} = \sum_{i=1}^{m} \text{ChannelReorder}^{-1}(g^{(i)}, \text{Compact}(\mathbf{W}^{(i)}))$. Here Compact reshapes the input matrix for the preparation of reordering its channels. It is easy to verify that $\text{ChannelReorder}^{-1}(g^{(i)}, \bullet) = \text{ChannelReorder}(\frac{D}{g^{(i)}}, \bullet)$.

2. Loading-inference consolidation. After loading the merged sparse weight matrix and merged bias matrix to memory, we can directly add them back to the weight matrix and bias matrix of the original FC layer. Formally, we use $\hat{\mathbf{W}}$ and $\hat{\mathbf{b}}$ to denote the final weight and bias of the FC layer for inference. Then $\hat{\mathbf{W}} = \mathbf{W} + \widetilde{\mathbf{W}}$, $\hat{\mathbf{b}} = \widetilde{\mathbf{b}}$. In this way, no additional inference cost is brought.

Overall, our consolidator reduces the storage space by using grouped connected layers and consolidating some duplicate entries. The training time non-linearity, *e.g.*, droppath, which turns out to be linear in inference time, effectively enriches model capacity under a constrained storage budget. Finally, we can consolidate the task-specific knowledge into the backbone model by merging the inference time linear components to enjoy free, efficient, and effective transfer learning.

Table 1: Full results on the VTAB-1k (Zhai et al., 2019) benchmark. The bold font denotes the best accuracy and the underline font denotes the second best accuracy in each column. Consolidator gives the strongest results, surpasses full fine-tuning by 7.56 accuracy on average, and outperforms the state-of-the-art methods with low storage overhead and no inference cost.

| | # params | Natural | | | | | | | Specialized | | | | Structured | | | | | | | | Average |
| | | Cifar100 | Caltech101 | DTD | Flowers102 | Pets | SVHN | Sun397 | Camelyon | EuroSAT | Resisc45 | Retinopathy | Clevr-Count | Clevr-Dist | DMLAB | KITTI-Dist | dSpr-Loc | dSpr-Ori | sNORB-Azim | sNORB-Ele | |
|---|---|---|---|---|---|---|---|---|---|---|---|---|---|---|---|---|---|---|---|---|---|
| Full | 100% | 68.9 | 87.7 | 64.3 | 97.2 | 86.9 | 87.4 | 38.8 | 79.7 | 95.7 | 84.2 | 73.9 | 56.3 | 58.6 | 41.7 | 65.5 | 57.5 | 46.7 | 25.7 | 29.1 | 68.97 |
| Head | 0.04% | 63.4 | 85.0 | 63.2 | 97.0 | 86.3 | 36.6 | 51.0 | 78.5 | 87.5 | 68.6 | 74.0 | 34.3 | 30.6 | 33.2 | 55.4 | 12.5 | 20.0 | 9.6 | 19.2 | 57.64 |
| Bias | 0.10% | 72.8 | 87.0 | 59.2 | 97.5 | 85.3 | 59.9 | 51.4 | 78.7 | 91.6 | 72.9 | 69.8 | 61.5 | 55.6 | 32.4 | 55.9 | 66.6 | 40.0 | 15.7 | 25.1 | 65.22 |
| VPT | 0.75% | **78.8** | 90.8 | 65.8 | 98.0 | 88.3 | 78.1 | 49.6 | 81.8 | **96.1** | 83.4 | 68.4 | 68.5 | 60.0 | 46.5 | 72.8 | 73.6 | 47.9 | **32.9** | 37.8 | 71.97 |
| Adapter | 0.36% | 69.8 | 91.2 | 68.8 | 99.0 | 89.9 | 85.7 | 53.8 | 82.3 | 95.5 | 83.7 | **76.1** | 82.4 | 65.0 | 48.2 | 80.5 | 74.5 | 49.7 | 29.8 | 39.2 | 74.28 |
| LoRA | 0.34% | 68.1 | 90.1 | 69.8 | 98.9 | 90.8 | 84.8 | 54.0 | 83.2 | 95.6 | 84.2 | 73.9 | 82.4 | 68.7 | 49.4 | 80.0 | 81.7 | 46.1 | 31.4 | 41.8 | 74.64 |
| AdaptFormer | 0.36% | 71.0 | 91.1 | 69.9 | 99.3 | 90.5 | 87.4 | 54.8 | 84.1 | 95.9 | 85.9 | 75.8 | **83.1** | 63.8 | 49.6 | 79.6 | 76.5 | 45.1 | 30.9 | 39.2 | 74.82 |
| NOAH | 0.52% | 69.6 | **92.7** | 70.2 | 99.1 | 90.4 | 86.1 | 53.7 | 84.4 | 95.4 | 83.9 | 75.8 | 82.8 | 68.9 | 49.9 | 81.7 | 81.8 | 48.3 | 32.8 | 44.2 | 75.48 |
| SSF | 0.29% | 69.0 | 92.6 | 75.1 | 99.4 | 91.8 | 90.2 | 52.9 | 87.4 | 95.9 | 87.4 | 75.5 | 75.9 | 62.3 | 53.3 | 80.6 | 77.3 | 54.9 | 29.5 | 37.9 | 75.69 |
| Ours | 0.35% | 74.2 | 90.9 | 73.9 | 99.4 | 91.6 | 91.5 | 55.5 | 86.9 | 95.7 | 86.6 | 75.9 | 81.2 | 68.2 | 51.6 | 83.5 | 79.8 | 52.3 | 31.9 | 38.5 | **76.53** |

## 4 EXPERIMENTS

### 4.1 EXPERIMENTAL SETTINGS

**Baselines.** We select several state-of-the-art parameter-efficient methods as our baselines, including Full, Head, Bias (Zaken et al., 2021), Adapter (Houlsby et al., 2019), VPT (Jia et al., 2022), LoRA (Hu et al., 2021), AdaptFormer (Chen et al., 2022), NOAH (Zhang et al., 2022) and SSF (Lian et al., 2022b). Note that Adapter, AdaptFormer, VPT, and NOAH will bring heavy inference cost to the pre-trained model which may cause troubles in resource-limited devices, while LoRA, SSF and our consolidator are more friendly for deployment with no extra inference cost brought.

**VTAB-1k.** We first run experiments on VTAB-1k (Zhai et al., 2019) benchmark, which covers a wide range of visual domains in 19 datasets. Each dataset contains 1000 images picked from the original dataset for training, and the size of test set stays unchanged, varying from 711 to 73728.

**Full data setting.** VTAB-1k merely has a small number of training images, and thus the capacity of a huge transformer is redundant to some extent. Therefore, to check the performance of parameter-efficient methods in a data-sufficient situation, we select 10 widely-used datasets for visual recognition in various domains and use the original training, validation, and test split for experiments. On average, a dataset in this setting contains a lot more images for training, leaving huge space for the model to adapt. The chosen datasets include natural pictures (Caltech101 (Fei-Fei et al., 2004), Cifar10 (Krizhevsky et al., 2009), Cifar100 (Krizhevsky et al., 2009)), fine-grained classification (CUB200 (Wah et al., 2011), OxfordFlowers (Nilsback & Zisserman, 2008), OxfordPets (Parkhi et al., 2012), StanfordDogs (Khosla et al., 2011)), textures (DTD (Cimpoi et al., 2014)), scene classification (SUN397 (Xiao et al., 2010)) and satellite images (EuroSAT (Helber et al., 2019)).

### 4.2 MAIN RESULTS

We first choose a ViT-B (Dosovitskiy et al., 2020) with 86M parameters as a base model.

**VTAB-1k** Tab. 1 presents the full results on VTAB-1k benchmark. Overall, our consolidator is the best parameter-efficient method. On 12 of 19 datasets, consolidator achieves the best or second best top-1 accuracy. Notably, consolidator surpasses the state-of-the-art methods NOAH and SSF by a clear margin, with low storage overhead and no inference cost.

**Full data setting** Tab. 2 presents the full results on full data setting. Overall, our consolidator still performs best. An interesting observation is that the rank of full fine-tuning rises as the training data increase. None of the parameter-efficient methods can reach comparable performance with full tine-tuning other than our consolidator within 0.5% parameter storage overhead. In contrast, the parameter-efficient methods can reach at least 5% higher accuracy on VTAB-1k than full fine-tuning under comparable or even lower storage budget (around 0.5%), as shown in Tab. 1.

Table 2: Full results on data-sufficient scenarios. It is more challenging to take full advantage of a large amount of data within a fairly small number of stored parameters. Consolidator earns the best or second best in all 10 datasets.

| | # params | Caltech101 | Cifar10 | Cifar100 | CUB | DTD | Flowers102 | Pets | Dogs | Sun397 | EuroSAT | Average |
|---|---|---|---|---|---|---|---|---|---|---|---|---|
| Full | 100% | 90.9 | 99.1 | **92.6** | **87.7** | **75.6** | 98.9 | **93.9** | 84.8 | 70.9 | **98.6** | 89.30 |
| Head | 0.10% | 90.0 | 88.8 | 71.6 | 83.5 | 71.3 | 98.2 | 90.5 | 78.4 | 67.0 | 95.6 | 83.49 |
| Bias | 0.22% | 90.2 | 99.0 | 91.9 | 86.4 | 73.5 | 98.5 | 92.6 | 85.4 | 71.0 | 97.7 | 88.62 |
| Adapter | 0.55% | 91.0 | 99.0 | 92.2 | 86.3 | 72.6 | 98.7 | 92.5 | 85.9 | 71.9 | 97.4 | 88.75 |
| LoRA | 0.53% | 90.1 | 99.0 | 91.8 | 85.4 | 71.9 | 98.1 | 92.1 | 85.6 | 71.3 | 97.2 | 88.25 |
| AdaptFormer | 0.55% | 90.9 | 99.0 | 92.3 | 86.5 | 73.4 | 98.9 | 92.9 | 85.2 | 71.4 | 97.1 | 88.76 |
| SSF | 0.30% | 90.9 | 99.0 | 92.1 | 85.7 | 72.6 | 98.3 | 92.3 | 85.7 | 71.6 | 97.8 | 88.60 |
| Ours | 0.50% | **91.4** | **99.1** | 92.5 | 87.0 | 74.5 | **99.0** | 93.2 | **86.4** | **72.1** | 97.9 | **89.31** |

Table 3: Adaptation results for a self-supervised model, MoCo v3 ViT-B. Consolidator earns the best average accuracy

| | VTAB-1k | | Full data | |
|---|---|---|---|---|
| | # params | Average | # params | Average |
| Full | 100% | 69.55 | 100% | 86.24 |
| Head | 0.04% | 59.62 | 0.10% | 75.42 |
| Bias | 0.10% | 69.15 | 0.22% | 80.96 |
| Adapter | 0.36% | 73.22 | 0.55% | 86.15 |
| LoRA | 0.34% | 72.73 | 0.53% | 79.70 |
| AdaptFormer | 0.36% | 74.03 | 0.55% | 86.26 |
| NOAH | 0.42% | 73.55 | —— | —— |
| SSF | 0.29% | 51.41 | 0.30% | 80.73 |
| Ours | 0.35% | **74.71** | 0.50% | **86.41** |

Table 4: Adaptation performance for more models in full data setting. Consolidator consistently reaches a better result than full fine-tuning within a very small set of parameters which is needed to be stored. Generally, it is more difficult to reach a better result than full fine-tuning when the model capacity is insufficient, *i.e.*, the model does not accommodate enough parameters in total.

| Architecture | | Supervised Learning | | | | | MAE |
|---|---|---|---|---|---|---|---|
| | | ViT-S | ViT-L | Swin-B | AS-MLP-B | ConvNeXt-B | ViT-B |
| Total Parameters | | 22M | 303M | 87M | 87M | 88M | 86M |
| Full | Average | 89.10 | 90.07 | 91.00 | **87.75** | **91.92** | 82.77 |
| | # params | 100% | 100% | 100% | 100% | 100% | 100% |
| Head | Average | 81.87 | 87.92 | 90.45 | 83.49 | 90.62 | 58.10 |
| | # params | 0.20% | 0.04% | 0.13% | 0.13% | 0.13% | 0.10% |
| Bias | Average | 87.86 | 89.81 | 91.00 | 86.26 | 91.61 | 79.80 |
| | # params | 0.44% | 0.13% | 0.36% | 0.32% | 0.28% | 0.22% |
| LoRA | Average | 87.55 | 89.86 | 90.95 | – | – | 82.22 |
| | # params | 5.12% | 0.33% | 0.80% | | | 1.82% |
| Adapter | Average | 88.28 | 89.91 | 90.98 | – | – | 82.89 |
| | # params | 5.09% | 0.35% | 0.78% | | | 1.80% |
| Ours | Average | **89.12** | **90.52** | **91.28** | 86.71 | 91.79 | **83.32** |
| | # params | 5.06% | 0.33% | 0.77% | 1.13% | 1.04% | 1.78% |

## 4.3 RESULTS FOR SELF-SUPERVISED VISION TRANSFORMER

Then we use a self-supervised trained transformer by MoCov3 (He et al., 2020) as our target model. Seen from Tab. 3, on both VTAB-1k and full data setting, consolidator consistently reaches the highest average top-1 accuracy. AdaptFormer gives the second highest accuracy. SSF significantly falls behind others (51.41 on VTAB and 80.73 on full data setting) when dealing with MoCo v3 ViT-B, showing limited generalization ability for self-supervised visual models.

## 4.4 RESULTS FOR MORE PRE-TRAINED MODELS

To further verify the generalization ability of consolidator, we conduct extensive experiments in Tab. 4 based on full data setting. First, we apply consolidator to supervised learned models with larger (ViT-L) or smaller (ViT-S) size than standard ViT-B. Compared with full fine-tuning, we achieve a comparable performance for ViT-S with 5.06% parameters. For ViT-L, merely 0.33% parameters can lead to 0.45 higher than full fine-tuning. Then we experiment on Swin-B, a hierarchical architecture using shifted windows to introduce locality for better recognition. We observe a 0.28 improvement while storing only 0.77% parameters. Next, we further verify the effectiveness of consolidator on other vision architectures other than transformers, *e.g.* AS-MLP (Lian et al., 2022a) and ConvNeXt (Liu et al., 2022). Finally, we experiment on ViT-B pre-trained by a generative SSL

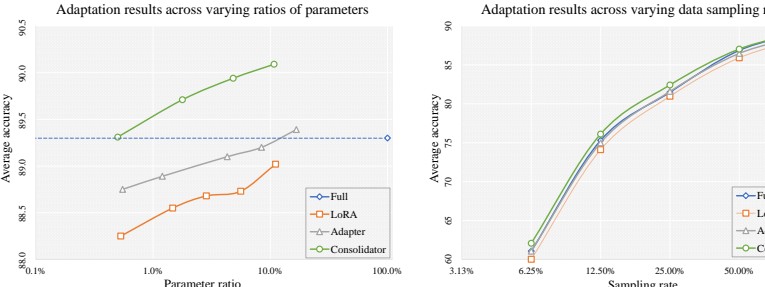

Figure 2: Left: the Adaptation results corresponding to varying ratios of stored parameters. Clearly, consolidator consistently outperforms LoRA, adapter, and full fine-tuning by a significant margin across a wide range of parameter scales from 0.5% to 10%. And each of the three methods will reach higher accuracy if we increase its storage budget and tune more parameters.    Right: adaptation results corresponding to varying sampling rates of data. Consolidator performs best across all sampling rates. As the sampling rate decreases, full fine-tuning slightly falls off its advantage over adapter and LoRA, which is consistent with our previous observations.

method, MAE, as a comparison with the contrastive SSL method MoCo v3. Our method stores 1.78% parameters onto the disk while enjoying 0.55 higher accuracy.

Generally, it is more difficult to perfom better than full fine-tuning when the model does not have enough parameters to fully leverage the information from massive training data.

## 4.5    ADAPTATION ACROSS VARYING SCALES OF STORED PARAMETERS

Next, we seek to find the principle between downstream accuracy and the number of stored parameters for all parameter-efficient methods with flexible parameter scales, based on full data setting. Results are shown in Fig. 2 left. LoRA, adapter, and consolidator reach better performance as the number of parameters increases. In various parameter scales (from 0.5% to 10%), our consolidator consistently outperforms all competitors by a clear margin.

## 4.6    ADAPTATION ACROSS VARYING DATA SAMPLING RATIOS

Fig. 2 right shows adaptation results corresponding to varying sampling rates of datasets, based on full data setting. For all sampling rates, consolidator keeps the best adaptation ability. In addition, as the sampling rate decreases, full fine-tuning slightly falls off its advantage over Adapter and LoRA, which is consistent with previous observations on VTAB-1k and full data setting.

## 4.7    RESULTS ON OBJECT DETECTION AND SEMANTIC SEGMENTATION.

We further verify our method on downstream object detection and semantic segmentation tasks. We adopt Swin-Base as the backbone which can provide hierarchical features. Experiments are done on PASCAL VOC 07+12 (Everingham et al., 2010) for detection and PASCAL VOC 12 (Everingham et al., 2010) for segmentation. We adopt Faster R-CNN (Ren et al., 2015) and UperNet (Xiao et al., 2018) separately for detection and segmentation framework. Seen from Tab. 5, our consolidator significantly outperforms full training and detection/segmentation head training with a small number of parameters stored, showing great potential for broader usage.

## 4.8    ABLATION STUDIES

We do controlled experiments to identify the effect of individual components in our module design. We report tuned parameters, stored parameters, and accuracy in Tab. 6. We experiment on 5 datasets with different domains: Caltech101, DTD, OxfordFlowers, StanfordDogs, and EuroSAT.

**Droppath *v.s.* Dropout.** We first investigate our choice of Droppath. Droppath with a consolidator of ($g^{(1)} = 96$, $g^{(2)} = 192$) as the base model. As shown in Tab. 6, compared with dropout and no drop-type layer, droppath can obtain 0.44 and 0.48 performance improvement, respectively, well demonstrating the effectiveness of encouraging stochastic depth for consolidator.

Table 5: Performance on downstream object detection and semantic segmentation tasks. Compared with tuning detection/segmentation head only, consolidator reaches much better mAP/mIoU with negligible parameters increase and surpasses the performance of tuning all parameters in both backbone and task head as well.

| Method | Backbone | Object Detection | | | Semantic Segmentation | | |
|---|---|---|---|---|---|---|---|
| | | Framework | # params | mAP | Framework | # params | mIoU |
| Full | | | 100% | 85.85 | | 100% | 83.24 |
| Head | Swin-B | Faster-RCNN | 16.74% | 84.89 | UperNet | 28.48% | 82.31 |
| **Consolidator** | | | **17.22%** | **86.64** | | **28.89%** | **83.96** |

Table 6: Effect of individual designs for a consolidator module.

| Branches | Original Bias | Extra Bias | Channel Reorder | Dropout | Droppath | Tuned Params | stored Params | Accuracy | $\Delta$Accuracy |
|---|---|---|---|---|---|---|---|---|---|
| 96, 192 | ✔ | ✔ | ✔ | ✗ | ✔ | 1.95% | 1.75% | 87.36 | -0.00 |
| 96, 192 | ✔ | ✔ | ✔ | ✔ | ✗ | 1.95% | 1.75% | 86.92 | -0.44 |
| 96, 192 | ✔ | ✔ | ✔ | ✗ | ✗ | 1.95% | 1.75% | 86.88 | -0.48 |
| 96, 192 | ✔ | ✔ | ✗ | ✗ | ✔ | 1.95% | 1.24% | 86.98 | -0.38 |
| 96, 192 | ✗ | ✔ | ✔ | ✗ | ✔ | 1.85% | 1.75% | 87.26 | -0.10 |
| 96, 192 | ✔ | ✗ | ✔ | ✗ | ✔ | 1.76% | 1.75% | 86.92 | -0.44 |
| 96, 192 | ✗ | ✗ | ✔ | ✗ | ✔ | 1.66% | 1.65% | 86.82 | -0.54 |
| 384 | ✔ | ✔ | ✔ | ✗ | ✔ | 0.56% | 0.47% | 87.08 | -0.00 |
| 384 | ✔ | ✔ | ✗ | ✗ | ✔ | 0.56% | 0.47% | 86.98 | -0.10 |
| 384, 384 | ✔ | ✔ | ✔ | ✗ | ✔ | 0.92% | 0.47% | 86.96 | -0.12 |
| 384 | ✗ | ✗ | ✔ | ✗ | ✔ | 0.37% | 0.37% | 86.84 | -0.00 |
| 384 (unst) | ✗ | ✗ | N/A | ✗ | ✔ | 0.37% | 0.37% | 86.70 | -0.14 |

**ChannelReorder.** The effect of ChannelReorder operation mainly lies in separating the entries into different branches to reduce the repetitive ones. In a multi-branch case like ($g^{(1)} = 96, g^{(2)} = 192$), ChannelReorder brings 0.38 accuracy improvement. Furthermore, it is helpful even if there is only one branch like ($g^{(1)} = 384$), where ChannelReorder still slightly raises the accuracy by 0.1.

**Duplication of bias and weight.** Based on the consolidator with ($g^{(1)} = 96, g^{(2)} = 192$), we can see duplicating bias is relatively effective. Compared with tuning the original bias and only tuning two extra biases, tuning all three biases can lead to 0.44 and 0.1 performance improvement, respectively, with the same storage cost. Additionally, we also compare tuning original bias *v.s.* not tuning bias, and the former only has a slight 0.1 accuracy advantage, which further verifies the effectiveness owes mostly to the delicate consolidation design instead of simple bias tuning. Besides bias, we test on a consolidator with ($g^{(1)} = 96$) to investigate the effect of integrating duplicate weights. However, this kind of consolidation does not bring notable improvement.

**Structured *v.s.* Unstructured.** One potential limitation of consolidator is that $g^{(i)}$ can not be selected arbitrarily for it must be a factor of the channels. This may cause trouble when fairly few parameters other than head parameters, *e.g.* 0.0001%, are required to be tuned. A solution is to adopt unstructured sparsity instead of structured block-wise sparsity in consolidator branches to flexibly control the parameter number. We simulate the situation with ($g^{(1)} = 384$) for comparison with a unstructured implementation. Seen from Tab. 6, the unstructured branch whose tunable parameter number equals that of a branch with $g = 384$ faces a slight accuracy drop, 0.14. In summary, the unstructured consolidator can be a sub-optimal choice when needed, with a slight performance drop and more training cost due to the unstructured sparse matrix being unfriendly to the hardware.

## 5 CONCLUSIONS

We propose consolidator, a novel method to achieve both parameter- and inference-efficient visual adaptation. Consolidator adds a few tunable, mergeable modules along each fully connected layer in the pre-trained model and keeps most of the original parameters frozen during adaptation. We design a two-stage consolidation to dramatically boost performance under a given storage budget. The duplicate entries in a consolidator will be merged into a single matrix and stored on disk. Finally, we consolidate the task-specific parameters in consolidator into the tasks-agnostic parameters in the pre-trained model, bringing no extra inference cost. On various tasks, consolidator outperforms all state-of-the-art competitors significantly, showing strong scalability and generalization ability.

ACKNOWLEDGMENTS

This work was supported by National Key R&D Program of China (No. 2021ZD0114703), National Natural Science Foundation of China (Nos. 61925107, 62271281, U1936202, 61571269), Beijing Natural Science Foundation (No. L223023), China Postdoctoral Science Foundation (BX2021161) and Zhejiang Lab Open Research Project (NO. K2022KI0AB01).

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

## A    EXPERIMENT SETTINGS

### A.1    IMAGE RECOGNITION

#### A.1.1    DATASETS

**VTAB-1k.** VTAB-1k (Zhai et al., 2019) consists of 19 visual datasets in three groups: Natural, Specialized, and Structured, containing images collected from a wide range of visual domains. Each dataset is divided into the training set (800 images), validation set (200 images), and test set (the original set). The final adaptation accuracy is reported on the test set. All models are fine-tuned on the full 1000 labeled images, *i.e.*, train+val set. The validation set is used for tuning some hyperparameters.

**Full data.** We select 10 widely-used visual datasets with sufficient data to further verify the effectiveness of the parameter-efficient methods. The datasets are shown in the following part:

*Caltech101* (Fei-Fei et al., 2004) contains pictures of natural objects belonging to 101 classes, with 3060 images for training and 6084 images for testing.

*CIFAR-10/100* (Krizhevsky et al., 2009) contains pictures of natural objects belonging to 10/100 classes, with 50000 images for training and 10000 images for testing.

*Caltech-UCSD Birds-200-2011 (CUB)* (Wah et al., 2011) contains pictures of birds belonging to 200 fine-grained bird classes, with 5994 images for training and 5794 images for testing.

*Describable Textures Dataset (DTD)* (Cimpoi et al., 2014) contains texture images in the wild belonging to 47 human-centric classes, with 5640 images in total which are equally split into the training, validation, and test set.

*OxfordFlowers102* (Nilsback & Zisserman, 2008) contains pictures of flowers belonging to 102 fine-grained flower classes, with 1020 images for training, 1020 images for validating, and 6149 images for testing.

*Oxford-IIITPets* (Parkhi et al., 2012) contains pictures of pets belonging to 37 fine-grained pet classes, with 3680 images for training and 3669 images for testing.

*StanfordDogs* (Khosla et al., 2011) contains pictures of dogs belonging to 120 fine-grained dog classes, with 12000 images for training and 8580 images for testing.

*Scene UNderstanding (SUN397)* (Xiao et al., 2010) contains pictures of scenes belonging to 397 classes, with 76128 images for training, 10875 images for validating, and 21750 images for testing.

*EuroSAT* (Helber et al., 2019) contains pictures based on Sentinel-2 satellite images belonging to 10 classes, with 27000 images in total.

Some of the datasets do not contain a validation split, and we will manually select ˜10% random images from the training set as the validation set.

To better utilize the massive data in various domains, we follow the practice in (Junguang Jiang & Long, 2020; Jiang et al., 2022) to make preparation and divide data splits. For data augmentation, we adopt a standard pipeline. In training, we do a random resize crop to 224×224, random horizontal flip, and normalization for each input image. In test, we do a resize to 256×256, center crop to 224×224, and normalization for each input image.

### A.1.2 TRAINING HYPERPARAMETERS.

On **VTAB-1k**, we follow the hyperparameters in VPT (Jia et al., 2022) for full fine-tuning, Head, Bias, and VPT, and mainly follow the hyperparameters in NOAH (Zhang et al., 2022) and SSF (Lian et al., 2022b) for adapter, LoRA, NOAH, and consolidator. The detailed hyperparameters for each tuning method can be found in Tab. 7.

On **full data setting**, we do a quick grid search to choose a proper set of training hyperparameters based on the performance of full fine-tuning for every well-trained visual representation. All training hyperparameters are shown in Tabs. 7 and 8. Following the practice in (Jiang et al., 2022; Junguang Jiang & Long, 2020), we reduce the learning rate of backbone parameters to 0.1x that of head parameters. We then adopt the same hyperparameters for all parameter-efficient variants as well as our consolidator after the hyperparameters have been chosen according to full fine-tuning results.

### A.1.3 METHODS.

*Full*: Ordinary fine-tuning. It tunes all parameters and stores all parameters to disk for every downstream task, leading to heavy storage cost.

*Head*: Also known as linear probing. It only tunes the classification head and freezes other parameters.

*Bias* (Zaken et al., 2021): Bias only tunes the biases and freezes all the weights in a pre-trained model.

*Adapter* (Houlsby et al., 2019): Adapter inserts a sequence of tunable parameters, including a down projection, a non-linearity (here we use GELU (Hendrycks & Gimpel, 2016)) and an up projection layer, into each encoder block. The adapters are serially connected with the backbone layers.

*LoRA* (Hu et al., 2021): LoRA adds a concurrent branch containing low-rank weight matrices for efficient parameter update. A LoRA module contains serial down projection and up projection layers without non-linearity, and thus it can be merged into the backbone parameters before inference.

*VPT* (Li & Liang, 2021; Jia et al., 2022): VPT appends tunable virtual tokens to the inputs of transformer blocks, which will participate in the calculation of the subsequent blocks along with the actual tokens.

*NOAH* (Zhang et al., 2022): NOAH first trains a large supernet with all three modules, VPT, LoRA, and adapter, and then searches for the optimal configurations of each module for each layer using

the NAS algorithm introduced by AutoFormer (Chen et al., 2021). In the end, NOAH retrains the best subnet candidates to produce the final result.

*AdaptFormer* (Chen et al., 2022): AdaptFormer adopts parallel adapters (Houlsby et al., 2019) and a scale operation for each encoder block.

*SSF* (Lian et al., 2022b): SSF adds tunable scale and shift parameters for each operation of the backbone model. The added parameters can be merged into the original model and thus SSF brings no inference cost.

**Implementation details.** On **VTAB-1k**, we follow the same implementations for LoRA and adapter with NOAH (Zhang et al., 2022).

On **Full data setting**, the detailed implementations are shown as follows. For LoRA (Hu et al., 2021), we follow its original implementation to do low-rank re-parameterization and tune merely two weight matrices $\mathbf{W}_q$ and $\mathbf{W}_v$, which generate the attention matrices $\mathbf{Q}$ and $\mathbf{V}$, in every transformer encoder block. For bias, we tune all the biases (including the parameters outside encoder blocks) in the model. For adapter, we follow its original implementation to additionally tune the parameters of LayerNorm (Ba et al., 2016) and add the tunable adapter modules to the end of each MHSA and MLP before calculating the residual connections. For consolidator, we follow the practice of adapter to tune the LayerNorm as well and only add consolidator to the linear layers in MHSA and MLP of each transformer encoder block. We sweep the drop ratio in {0.0, 0.2, 0.5, 0.8}. We choose training hyperparameters according to the performance of full-finetuning for each visual representation. The hyperparameters are finally configured as in Tab. 7. When applying consolidator to Swin-B, we skip the first stage of transformer encoder blocks that contain a relatively small part of parameters (0.6% of total parameters) compared with others and merely insert consolidator for linear layers in MHSA and MLP of blocks of last 3 stages.

## A.2 OBJECT DETECTION

On the downstream object detection task, we adopt Faster R-CNN (Ren et al., 2015) framework with FPN (Lin et al., 2017) to verify the effectiveness of consolidator on Pascal VOC 07+12 dataset (Everingham et al., 2010). We use a Swin-B pre-trained on IN-21k as the backbone model. The consolidator setting is the same as the setting of Swin-B in Tab. 7. For hyperparameters, we adopt AdamW as the optimizer with a learning rate of 1e-4 and weight decay of 5e-2, and train for 8 epochs in total. The learning rate is decayed by a factor of 10 after 6 epochs. The first 300 iterations are trained with a warmup ratio of 1e-3 for the learning rate. The data augmentation is the same as the default strategy in mmdetection (Chen et al., 2019).

## A.3 SEMANTIC SEGMENTATION

On the downstream semantic segmentation task, we adopt UperNet (Xiao et al., 2018) to verify the effectiveness of consolidator on Pascal VOC 12 dataset (Everingham et al., 2010). We use a Swin-B pre-trained on IN-21k as the backbone model. The consolidator setting is the same as the setting of Swin-B in Tab. 7. For hyperparameters, we adopt AdamW as the optimizer with a learning rate of 6e-5 and weight decay of 1e-2, and train for 20000 iterations in total. The learning rate is decayed by a polynomial scheduler with a power of 1.0. The first 200 iterations are trained with a warmup ratio of 1e-6 for the learning rate. The data augmentation is the same as the default strategy in mmsegmentation (Contributors, 2020).

## B TRAINING AND INFERENCE COST

Many the classic parameter-efficient tuning methods, *e.g.* adapter (Houlsby et al., 2019) and Adaptformer (Chen et al., 2022) introduce non-negligible extra cost in inference period and thus slow down the processing speed. In contrast, consolidator tuning shares identical structure with the original model and bring no extra inference cost. We quantitatively show the training cost and inference cost across different parameter scales in Fig. 3. Here we do not show the cost of SSF for it can not be adapted to different scales of parameter budget. In addition, VPT and NOAH both search for an optimal structure from a large searching space and thus it is hard to fairly measure their cost as well.

Table 7: The method-related hyperparameters.

| | model | Setting | LoRA | Adapter | Consolidator |
|---|---|---|---|---|---|
| Supervised, MoCo v3 | ViT-B | VTAB-1k | rank=8 | hidden=16 | $(g^{(1)}{=}384)$ |
| | | Full data | rank=10 | hidden=9 | $(g^{(1)}{=}384)$ |
| Supervised | ViT-S | Full data | rank=58 | hidden=56 | $(g^{(1)}{=}48, g^{(2)}{=}64, g^{(3)}{=}96)$ |
| Supervised | ViT-L | Full data | rank=9 | hidden=8 | $(g^{(1)}{=}512)$ |
| Supervised | Swin-B | Full data | rank=12 | hidden=10 | $(g^{(1)}{=}256)$ |
| Supervised | AS-MLP-B | Full data | —— | —— | $(g^{(1)}{=}128)$ |
| Supervised | ConvNeXt-B | Full data | —— | —— | $(g^{(1)}{=}128)$ |
| MAE | ViT-B | Full data | rank=40 | hidden=38 | $(g^{(1)}{=}96, g^{(2)}{=}192)$ |

Table 8: The training hyperparameters on full data setting.

| | Supervised | | | | | MAE | MoCo v3 |
|---|---|---|---|---|---|---|---|
| model | ViT-S | ViT-L | Swin-B | AS-MLP-B | ConvNeXt-B | ViT-B | ViT-B |
| pretraining dataset | IN-21k | IN-21k | IN-21k | IN-1k | IN-21k | IN-1k | IN-1k |
| optimizer | sgd | sgd | sgd | sgd | sgd | sgd | sgd |
| warmup epochs | 5 | 5 | 5 | 5 | 5 | 5 | 5 |
| epochs | 40 | 40 | 40 | 40 | 40 | 40 | 40 |
| batch size | 128 | 32 | 64 | 48 | 64 | 64 | 64 |
| lr | 1e-3 | 1e-3 | 1e-3 | 1e-3 | 1e-3 | 1e-2 | 1e-2 |
| wd | 1e-4 | 1e-4 | 1e-4 | 1e-4 | 1e-4 | 1e-4 | 1e-4 |

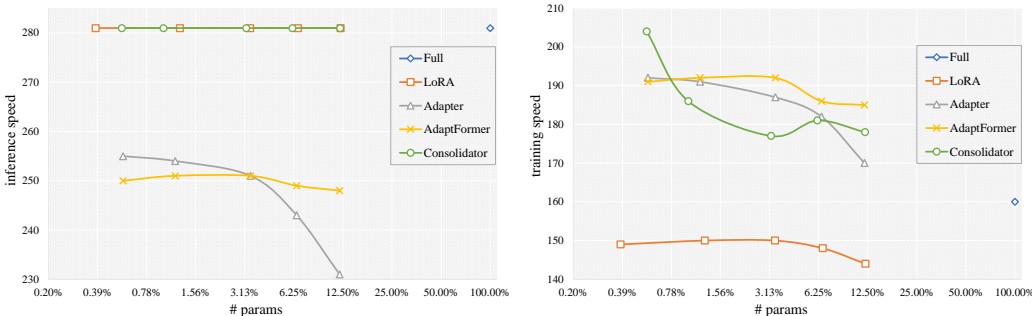

Figure 3: Left: comparison of the inference speed (images/second). Right: comparison of the training speed (images/second). We can conclude that consolidator tuning maintains good throughput during training across various storage budgets, and bring no extra cost compared with normal fine-tuning in inference period.

Table 9: Sensitivity test on consolidator's hyperparameters. Given a particular target storage budget, we may have several choices in selecting the branches and groups. As seen below, such choices make little influence on the final results. The performance of consolidator is relatively stable under a given storage budget.

| # params | 0.75% | 1.01% | 4.88% | 6.42% |
|---|---|---|---|---|
| branches$_{more}$ | (256,768) | (256,384,768) | (48,64,96) | (24,48) |
| acc$_{more}$ | 87.02 | 87.04 | 87.18 | 87.28 |
| branches$_{less}$ | (192) | (128) | (32,64) | (16) |
| acc$_{less}$ | 86.98 | 87.00 | 87.14 | 87.36 |
| $\Delta$acc | 0.04 | 0.04 | 0.04 | -0.08 |

## C    SENSITIVITY OF GROUPS AND BRANCHES IN CONSOLIDATOR

Given a particular target storage budget, we may have several choices in selecting the branches and groups. As seen in Tab. 9, such choices make little influence on the final results. The performance of consolidator is relatively stable under a given storage budget. When the budget increases, the performance of consolidator increases as well. Such monotonical property is helpful for real-world applications, making it easy to tune hyperparameters and rapidly find an optimal candidate under a given storage budget.

