# OpenReview forum: "Consolidator: Mergable Adapter with Group Connections for Visual Adaptation"
_ICLR.cc/2023/Conference — ICLR 2023 poster_

### Official Review · Reviewer_2eph · 2022-10-19

**Confidence:** 4
**Correctness:** 4
**Technical Novelty And Significance:** 2
**Empirical Novelty And Significance:** 3
**Recommendation:** 5

**Clarity, Quality, Novelty And Reproducibility:**

The paper is clearly written and easy to follow.

A typo in Sec 3.2: **aim ti**.

There is no source code available so I can not evaluate the reproducibility currently.


**Strength And Weaknesses:**

#### Strength

1. The design of the consolidator is parameter-efficient since its mergeable property. By extending a single branch to multiple branches, the consolidator has better flexibility and transferability.

2.  The two-stage process maximizes the adaptation capacity without extra inference costs.

3. The paper is well-written and easy to follow.

#### Weaknesses

1. The novelty of this work is limited. The mergeable design is proposed by LoRA [1] for efficient parameter finetuing. Besides, what's the main difference between the **Channel Reorder** and **shuffle** operator in ShuffleNet[2]?

2. Some recent related works are not discussed, e.g. [3][4][5].

3. The improvement is marginal compared with NOAH in Table 1.


[1] Hu, Edward J., et al. "Lora: Low-rank adaptation of large language models." arXiv 2021.

[2] Zhang, Xiangyu, et al. "Shufflenet: An extremely efficient convolutional neural network for mobile devices." CVPR 2018.

[3] Bahng, Hyojin, et al. "Exploring visual prompts for adapting large-scale models." arXiv 2022.

[4] Chen, Shoufa, et al. "AdaptFormer: Adapting Vision Transformers for Scalable Visual Recognition." NeurIPS 2022.

[5] Jie, Shibo, and Zhi-Hong Deng. "Convolutional bypasses are better vision transformer adapters." arXiv 2022.

**Summary Of The Paper:**

This work proposed **consolidator** to efficiently fine-tune transformers. Consolidator has multi-branches structures, which can be merged into a single matrix and saved on the disk. Extensive experiments demonstrate the efficiency and effectiveness of consolidator on various vision tasks.

**Summary Of The Review:**

See above.

---

> ### Author Response · Authors · 2022-11-17
> **Response for Reviewer 2eph**
>
> Thank you very much for pointing out the problems in detail for us to further improve our manuscript.
>
> **Question 1.1**: The novelty of this work is limited. The mergeable design is proposed by LoRA [1] for efficient parameter finetuing. Besides, what's the main difference between the **Channel Reorder** and **shuffle** operator in ShuffleNet[2]?
>
> **Answer**: Thanks for the question. The computation rules of shuffle operator and ChannelReorder are similar. The main difference is that the shuffle operator in ShuffleNet receives 4-dimensional inputs and generates 4-dimensional outputs, and ChannelReorder receives 3-dimensional inputs and generates 3-dimensional outputs.
>
> We agree with you that the mergeable structures are already proposed before our paper, e.g. LoRA, ACNet, RepVGG, as we have discussed in the Related Work section. We have to emphasize the main contributions of this paper again:
> 1. Motivated by the success of group-wise convolution in visual tasks, we assume the channel-wise connections between subsequent visual features are sparse, and thus design **grouped connected (GC) layer** to do **effective visual adaptation**. Further, we extend the single branch to the multi-branch topology for better flexibility.
> 2. Previous works like LoRA merely merge parameters in **loading-inference** phase. None of the existing methods merge parameters in **training-storage** phase. In contrast, we are the first to decouple the tunable parameters and stored parameters. We design a **two-stage consolidation** scheme by merging corresponding parameters in both **training-storage** phase and **loading-inference** phase. Such merging in **training-storage** benefits the model's performance a lot under a given storage budget because it can tune more parameters for adaptation. In this way, we can maximally dig the adaptation capacity of the model under a constrained storage budget, with no extra inference cost.
> 3. Our method clearly reaches state-of-the-art performance, compared with the most recent methods [4][6] published at NIPS'22 as shown in Tab. 1 in our revised paper.
>
> **Question 2 & 3**: Some recent related works are not discussed, e.g. [3][4][5]. The improvement is marginal compared with NOAH in Table 1.
>
> **Answer**: Thanks for your supplement. We have added discussions about [3][4][5][6] in Related Work section, and added the experiment results of two state-of-the-art methods published at NIPS'22, AdaptFormer[4] and SSF[6] for further comparisons. Under our previous training configuration, it is unfair to directly compare the performance of our consolidator and other methods like VPT, NOAH and SSF, for they searches for a much larger space for the selection of hyper-parameters during training. For fair comparison, we mostly follow the configuration of [6] to re-run our experiments and update the newest results in Tab. 1 of our revised paper. In summary, compared with those most recent state-of-the-art methods AdaptFormer (average accuracy: 74.82), NOAH (average accuracy: 75.48) and SSF (average accuracy: 75.69), our consolidator (average accuracy: **76.53**) outperforms their results by a clear margin. We also added such additional experiments for AdaptFormer and SSF in Tab.2 and Tab. 3 for further comparisons. Please refer to **Tab. 1, Tab. 2 and Tab. 3** for more details.
>
> Supervised ViT-B:
> | Method | VTAB-1k | Full data |
> | - | - | - |
> | AdaptFormer | 74.82 | 88.76 |
> | NOAH | 75.48 | - |
> | SSF | 75.69 | 88.60 |
> | Consolidator | **76.53** | **89.31** |
>
> MoCo v3 ViT-B:
> | Method | VTAB-1k | Full data |
> | - | - | - |
> | AdaptFormer | 74.03 | 86.26 |
> | NOAH | 73.55 | - |
> | SSF | 51.41 | 80.73 |
> | Consolidator | **74.71** | **86.41** |
>
> Specially, compared with NOAH, consolidator enjoys **1.05** accuracy gain with merely $\frac{0.35\%}{0.52\%}\approx 67$% storage cost. Moreover, NOAH brings heavy inference cost (latency and memory) for bringing extra structures and parameters, while our consolidator can be merged into the original structures and brings no extra cost.
>
> **Question 4**: A typo in Sec 3.2: **aim ti**.
>
> **Answer**: Thank you very much for figuring out our writing error in detail. We have fixed it: aim ti -> aim to. And we further review all the texts with the help of a typing assistant to find the potential remaining problems and revise our paper carefully.
>
> [1] Hu, Edward J., et al. "Lora: Low-rank adaptation of large language models." arXiv 2021.
>
> [2] Zhang, Xiangyu, et al. "Shufflenet: An extremely efficient convolutional neural network for mobile devices." CVPR 2018.
>
> [3] Bahng, Hyojin, et al. "Exploring visual prompts for adapting large-scale models." arXiv 2022.
>
> [4] Chen, Shoufa, et al. "AdaptFormer: Adapting Vision Transformers for Scalable Visual Recognition." NeurIPS 2022.
>
> [5] Jie, Shibo, and Zhi-Hong Deng. "Convolutional bypasses are better vision transformer adapters." arXiv 2022.
>
> [6] ‘Scaling & Shifting Your Features: A New Baseline for Efficient Model Tuning’, in NeurIPS2022.

---

> ### Author Response · Authors · 2022-12-11
> **Dear Reviewer 2eph**
>
> Thank you again for your time spent on this paper. We wonder if your concerns have been solved, and it would be great if you give us further feedback based on our revised version of the manuscript.

---

### Official Review · Reviewer_zLNL · 2022-10-24

**Confidence:** 4
**Clarity, Quality, Novelty And Reproducibility:** See Strength And Weaknesses
**Correctness:** 4
**Technical Novelty And Significance:** 3
**Empirical Novelty And Significance:** 3
**Recommendation:** 5

**Strength And Weaknesses:**

Strength:
1. Consolidator is a very effective method. Compared with other adapter methods and full parameter fine-tuning methods, Consolidator can achieve an almost optimal transfer learning effect.
2. Compared to the adapter methods, the design in this paper can be merged with the original model so that the inference phase does not increase the cost of memory and computation time.

Weakness:
1. The prior of ChannelReorder operation is strong but the explanation of its effectiveness is weak.
2. The parameters in all tables are stored parameters for Consolidator but not trainable parameters.
3. It should be possible to save time compared to the full parameter finetune during training, I wonder why this is not reported, nor is it compared to other baseline training time.
4. The value of saving storage space is not so great, especially for the base-level model, and there are few experiments on the Large model.
5. There are slightly more spelling errors and grammatical errors. P3 workss; P4 ti, thw; P5 the all the; P7 accomodate.


**Summary Of The Paper:**

This paper presents a new fine-tuning method for vision transformers that only trains and stores very few parameters. The fine-tuning method achieves both efficiency and effectiveness through its design of mergeable multiple branches and parameter shifting. Extensive experiments show the method surpasses other adapter methods and full fine-tuning on several transfer learning tasks.

**Summary Of The Review:**

See Strength And Weaknesses

---

> ### Author Response · Authors · 2022-11-17
> **Response for Reviewer zLNL**
>
> Thank you very much for pointing out the problems in detail for us to further improve our manuscript.
>
> **Question 1**: The prior of ChannelReorder operation is strong but the explanation of its effectiveness is weak.
>
> **Answer**: Thanks for the comment. The main purpose of ChannelReorder is to improve the flexibility of consolidator tuning, along with the multi-branch design. Our aim is to flexibly tune the number of parameters and enrich the information flow when each time a new branch is added. In Tab. 5, we have shown that the duplication of weight parameters in FC layers did not bring much improvement, but the duplication of bias parameters in FC layers was quite effective. Therefore, with the help of ChannelReorder, we can make the duplication entries on weight parameters less and effectively enrich the model's capacity by duplicating bias parameters, whenever a new branch is added to the module.
>
> **Question 2**: The parameters in all tables are stored parameters for Consolidator but not trainable parameters.
>
> **Answer**: Thanks for the interesting question. We would like to clarify that we did not show the trainable parameters in our tables because the trainable parameters are of little importance. The number of trainable parameters is not directly related to the training speed or memory cost (the structure of trainable module matters more), containing little information for the readers. Furthermore, it is usually acceptable to use more resources in training in most situations and people are more concerned about the cost when deploy the model to edge devices. In deployment situation, the stored parameters are exactly what we have to save on the disk, and thus the storage cost for each downstream task is proportional to the number of stored parameters.
>
> **Question 3**: It should be possible to save time compared to the full parameter finetune during training, I wonder why this is not reported, nor is it compared to other baseline training time.
>
> **Answer**: Thanks for the supplement. We have added the comparison of training speed between our method and existing state-of-the-art methods in **Fig. 3** in the revised paper. In summary, consolidator tuning maintains good throughput during training across various storage budgets.
>
> **Question 4**: The value of saving storage space is not so great, especially for the base-level model, and there are few experiments on the Large model.
>
> **Answer**: Thank you for the question!  As we have stated in the introduction section, we would like to emphasize the the application scenario of our consolidator from two aspects: **effective** adaptation and **efficient** adaptation, which are both important.
> 1.  **Effective**. Recently, the scale of vision models grows faster and faster. To sufficiently leverage the capacity of such huge model, a lot of training data are required, e.g. millions of or even more images. However, the downstream tasks in real-world usually do not contain such many data. Classic fine-tuning which tunes all parameters in a huge model easily falls into an overfitting situation, leading to inferior performance. Consolidator provides a powerful solution to take full advantage of the rich knowledge contained in the huge models. For example, as in Tab. 1, on VTAB benchmark, fine-tuning merely reaches 68.97% accuracy on average, while our consolidator can reach **76.53%** accuracy on average, **7.56%** higher than that of fine-tuning.
> 2. **Efficient**. In real-world scenarios, the DL models are usually trained/fine-tuned on a computation center, e.g. a powerful server with 8 NVIDIA V100 GPUs and sufficient disk space, and then deployed and doing inference on a resource-limited embedded device, e.g. a typical embedded AI device Jetson TX1[1] with merely 16GB storage, 1 GPU and 4GB GPU memory. We focus on the deployment phase. A ViT-L contains 307M 32-bit floating point parameters, and thus occupies 307 M $\times$ 32 bit $\div$ 8 bit/Byte $\div$ 1024 MB/GB $\approx$ 1.2GB storage space. Clearly, the storage space of such embedded devices is not sufficient especially considering that there are plenty of tasks to be adapted. Furthermore, we have shown in Tab. 4 that consolidator shows more capacity when compared with full fine-tuning for bigger models by comparing the performance and storage cost of ViT-S, ViT-B and ViT-L. Though we do not experiment on larger models like ViT-H due to the limited computation resources, we have no reason to doubt that consolidator can deal with bigger models with fantastic performance.
>
> **Question 5**: There are slightly more spelling errors and grammatical errors. P3 workss; P4 ti, thw; P5 the all the; P7 accomodate.
>
> **Answer**: Thank you very much for figuring out our writing errors in detail. We have fixed them: workss -> works, ti -> to, thw -> the, accomodate -> accommodate. And we further review all the texts with the help of a typing assistant to find the potential remaining problems and revise our paper carefully.

---

> ### Author Response · Authors · 2022-12-11
> **Dear Reviewer zLNL**
>
> Thank you again for your time spent on this paper. We wonder if your concerns have been solved, and it would be great if you give us further feedback based on our revised version of the manuscript.

---

### Official Review · Reviewer_ULMu · 2022-10-25

**Confidence:** 4
**Correctness:** 3
**Technical Novelty And Significance:** 3
**Empirical Novelty And Significance:** 3
**Recommendation:** 8

**Clarity, Quality, Novelty And Reproducibility:**

It should not be hard to reproduce this work because the authors describe the specific details for training and hyper parameters. The authors also propose consolidation process, which is also original in efficient transfer learning.

**Strength And Weaknesses:**

Strength:

(1) The authors propose a basic module, a consolidator for efficient transfer learning.

(2) A two-stage consolidation process is designed.

(3) Extensive experiments on various downstream tasks show state-of-the-art performance.

Weaknesses:
This work aims to design a mergeable adapter for efficient transfer learning and proposes a two-stage consolidation process by merging corresponding parameters. I have some questions as follows.

(1)  In Figure 1, the authors split channels and use zero padding, which might cause a heavy computational cost. Thus, the training time/speed should be shown for comparison with other tuning methods. Besides, I notice that the authors insert a consolidator in the FC layer in Figure 1. Will performance be further enhanced if the consolidator is inserted in other locations? Some recent tuning methods [1, 2] should also be discussed and compared in Table 1.

(2)  Some other tuning baselines are missing in Table 4. The authors should conduct more experiments with other tuning methods for fair comparisons.

(3)  The consolidator is designed for a linear layer. Can it be applied to other operational structures, such as self attention. Further, the current consolidator module is designed mainly in the vision transformer. In Table 1, Table 4 and Table 5, the proposed method obtains promising performance compared to the Full tuning. Push it forward, can it be extended to other model families such as CNNs or MLP architectures [3,4,5]? These will be interesting experiments. Due to the rebuttal time constraints, the authors do not need to add these extension experiments, but feel free to discuss them.

(4)  There exists a large number of typos in the current version, the authors are able to polish it further.

[1] ‘AdaptFormer: Adapting Vision Transformers for Scalable Visual Recognition’, in NeurIPS2022.

[2] ‘Scaling & Shifting Your Features: A New Baseline for Efficient Model Tuning’, in NeurIPS2022.

[3] ‘A ConvNet for the 2020s’, in CVPR2022.

[4] ‘AS-MLP: An Axial Shifted MLP Architecture for Vision’, in ICLR2022.

[5] ‘RepMLPNet: Hierarchical Vision MLP with Re-parameterized Locality’, in CVPR2022.


**Summary Of The Paper:**

This work proposes a consolidator that modifies the pretrained model with the addition of a small set of tunable parameters to store the task-specific knowledge while freezing the backbone model during adaptation. A two-stage consolidation process is designed by merging corresponding parameters in the training-storage phase and loading-inference phase. Extensive experiments are conducted on various downstream tasks and the results outperform state-of-the-art methods with fewer stored parameters but superior performance.

**Summary Of The Review:**

This work proposes a two-stage consolidation process for efficient transfer learning. It outperforms state-of-the-art methods with fewer stored parameters but superior performance. I tend to vote marginally above the acceptance threshold. Some additional experiments should be conducted to further improve the quality of this manuscript.

---

> ### Author Response · Authors · 2022-11-17
> **Response for Reviewer ULMu (part 1/2)**
>
> Thank you very much for your acknowledgement of our work and the kind suggestions.
>
> **Question 1.1**: In Figure 1, the authors split channels and use zero padding, which might cause a heavy computational cost. Thus, the training time/speed should be shown for comparison with other tuning methods.
>
> **Answer**: Thank you for the advice. We have to clarify that the formulations in the Methodology section aim to convey our idea and design accurately, and we do not really adopt zero padding to implement the GC layer in our code. The implementation is more efficient than the formulations. We have added the comparison of training speed between our method and existing state-of-the-art methods in the revised paper. Please refer to Fig. 3 for more details. In summary, consolidator tuning maintains good throughput during training across various storage budgets,
>
> **Question 1.2 & Question 3.1**: Besides, I notice that the authors insert a consolidator in the FC layer in Figure 1. Will performance be further enhanced if the consolidator is inserted in other locations? The consolidator is designed for a linear layer. Can it be applied to other operational structures, such as self attention.
>
> **Answer**: Thank you for your careful observation. In our current design, the consolidator has already attended the calculation of the self-attention module.
>
> A transformer encoder is composed by a Multi-Head Self-Attention (MHSA) and a Multi-Layer Perceptron (MLP). For MHSA, the input features are first processed by three FC layers to generate matrices $Q,K,V$, and the output is calculated by $Softmax(\frac{QK^T}{\sqrt{d}})V$ and then projected by another FC layer. Therefore, the parametric components of MHSA are four FC layers. Similarly, the parametric components of MLP are two FC layers as well. We formulate our consolidator for all the FC layers, which can already sufficiently cover all the parametric components in each MHSA and MLP.
>
> We have made it clearer in Section 3.1 in our revised paper.
>
> **Question 1.3**: Some recent tuning methods [1, 2] should also be discussed and compared in Table 1.
>
> **Answer**: Thanks for the supplement. In our revised paper, we have discussed AdaptFormer[1] and SSF[2] in the Related Work section and added the experimental results of them in Tab. 1. for comparisons. For fair comparisons, we mostly follow the experimental configuration of [2] to re-run the experiments on VTAB-1k of our proposed consolidator and update our newest results in the revised paper. In summary, compared with those most recent state-of-the-art methods AdaptFormer (average accuracy: 74.82) and SSF (average accuracy: 75.69), our consolidator (average accuracy: 76.53) still shows superior performance. Below is the additional results in Tab. 1. We also added the experimental results of AdaptFormer and SSF in Tab. 2 and Tab. 3 for further comparisons. Please refer to our revised paper for more details.
>
> Supervised ViT-B:
> | Method | VTAB-1k | Full data |
> | - | - | - |
> | AdaptFormer | 74.82 | 88.76 |
> | NOAH | 75.48 | - |
> | SSF | 75.69 | 88.60 |
> | Consolidator | **76.53** | **89.31** |
>
> MoCo v3 ViT-B:
> | Method | VTAB-1k | Full data |
> | - | - | - |
> | AdaptFormer | 74.03 | 86.26 |
> | NOAH | 73.55 | - |
> | SSF | 51.41 | 80.73 |
> | Consolidator | **74.71** | **86.41** |
>
> [1] ‘AdaptFormer: Adapting Vision Transformers for Scalable Visual Recognition’, in NeurIPS2022.
>
> [2] ‘Scaling & Shifting Your Features: A New Baseline for Efficient Model Tuning’, in NeurIPS2022.
>
> [3] ‘A ConvNet for the 2020s’, in CVPR2022.
>
> [4] ‘AS-MLP: An Axial Shifted MLP Architecture for Vision’, in ICLR2022.
>
> [5] ‘RepMLPNet: Hierarchical Vision MLP with Re-parameterized Locality’, in CVPR2022.

---

> ### Author Response · Authors · 2022-11-17
> **Response for Reviewer ULMu (part 2/2)**
>
>
> **Question 2**: Some other tuning baselines are missing in Table 4. The authors should conduct more experiments with other tuning methods for fair comparisons.
>
> **Answer**: Thank you for the advice. In Tab. 4 of our revised paper, we add the results of head, bias, adapter and LoRA, for further comparisons. Please see our revised paper for more details. In short, our method still consistently outperforms other baselines.
>
> | architecture | method | #params | accuracy |
> | - | - | - | - |
> | ViT-S | head | 0.20% | 81.87 |
> | ViT-S | bias | 0.44% | 87.86 |
> | ViT-S | LoRA | 5.12% | 87.55 |
> | ViT-S | Adapter | 5.09% | 88.28 |
> | ViT-S | Consolidator | 5.06% | **89.12** |
> | ViT-L | head | 0.04% | 87.92 |
> | ViT-L | bias | 0.13% | 89.81 |
> | ViT-L | LoRA | 0.33% | 89.86 |
> | ViT-L | Adapter | 0.35% | 89.91 |
> | ViT-L | Consolidator | 0.33% | **90.52** |
> | Swin-B | head | 0.13% | 90.45 |
> | Swin-B | bias | 0.36% | 91.00 |
> | Swin-B | LoRA | 0.80% | 90.95 |
> | Swin-B | Adapter | 0.78% | 90.98 |
> | Swin-B | Consolidator | 0.77% | **91.28** |
> | MAE ViT-B | head | 0.10% | 58.10 |
> | MAE ViT-B | bias | 0.22% | 79.80 |
> | MAE ViT-B | LoRA | 1.82% | 82.22 |
> | MAE ViT-B | Adapter | 1.80% | 82.89 |
> | MAE ViT-B | Consolidator | 1.78% | **83.32** |
>
> **Question 3.2**: Further, the current consolidator module is designed mainly in the vision transformer. In Table 1, Table 4 and Table 5, the proposed method obtains promising performance compared to the Full tuning. Push it forward, can it be extended to other model families such as CNNs or MLP architectures [3,4,5]? These will be interesting experiments. Due to the rebuttal time constraints, the authors do not need to add these extension experiments, but feel free to discuss them.
>
> **Answer**: Yes, it is an interesting point we have ignored before. Due to limited time, we merely select one representative MLP, AS-MLP[4], and one representative CNN, ConvNeXt[3], to verify our method. Specially, we conduct experiments on the two new architectures with four tuning patterns, full, head, bias, and consolidator, and show the results in Tab. 4 in our revised paper. In summary, consolidator keeps competitive for MLP or CNN architecture. However, for AS-MLP, it takes more storage budget for our consolidator to reach the result of full fine-tuning, compared with ConvNeXt and transformers with similar number of parameters, 86M. Probably it is because the expression power of AS-MLP is slightly weak.
>
> | architecture | method | #params | accuracy |
> | - | - | - | - |
> | AS-MLP | full | 100% | **87.75** |
> | AS-MLP | head | 0.13% | 83.49 |
> | AS-MLP | bias | 0.32% | 86.26 |
> | AS-MLP | Consolidator | 1.13% | *86.71* |
> | ConvNeXt | full | 100% | **91.92** |
> | ConvNeXt | head | 0.13% | 90.62 |
> | ConvNeXt | bias | 0.28% | 91.61 |
> | ConvNeXt | Consolidator | 1.04% | *91.79* |
>
> [1] ‘AdaptFormer: Adapting Vision Transformers for Scalable Visual Recognition’, in NeurIPS2022.
>
> [2] ‘Scaling & Shifting Your Features: A New Baseline for Efficient Model Tuning’, in NeurIPS2022.
>
> [3] ‘A ConvNet for the 2020s’, in CVPR2022.
>
> [4] ‘AS-MLP: An Axial Shifted MLP Architecture for Vision’, in ICLR2022.
>
> [5] ‘RepMLPNet: Hierarchical Vision MLP with Re-parameterized Locality’, in CVPR2022.

---

> ### Author Response · Authors · 2022-12-11
> **Dear Reviewer ULMu**
>
> Thank you again for your time spent on this paper. We wonder if your concerns have been solved, and it would be great if you give us further feedback based on our revised version of the manuscript.

---

> > ### Comment · Reviewer_ULMu · 2022-12-13
> > **Response to the authors**
> >
> > Thanks for the authors' responses. Amounts of experiments are conducted in the rebuttal stage for comparisons. Additionally, the method section is also clarified, which addresses my concerns. I am willing to raise my rating to 'accept'.

---

### Official Review · Reviewer_HhUN · 2022-10-26

**Confidence:** 4
**Clarity, Quality, Novelty And Reproducibility:** I'm Ok with the quality, clarity, and…
**Correctness:** 3
**Technical Novelty And Significance:** 3
**Empirical Novelty And Significance:** 2
**Recommendation:** 5

**Strength And Weaknesses:**

+ Compared to the baselines and other methods, the proposed methods could bring some improvements in accuracy and computation cost.
+ Some ablation experiments are introduced to prove the effectiveness of the proposed method.

The main concerns are listed below.
- Strictly restricted application scenario. The author introduces the application scenario in the abstract where the resource-limited devices cannot store a full copy of parameters but could provide computation to train/fine-tune the large models. It is a bit strange the devices with the ability to train DL models do not have sufficient storage.
- The proposed method is irrelevant to Transformer. It is closer to the re-parameterization technology.
- Some recent literature on the adaptation of Transformers is missing.

[a] AdaptFormer: Adapting Vision Transformers for Scalable Visual Recognition, NeurIPS 2022.

**Summary Of The Paper:**

This paper studies the adaptation of the well-trained transformer to downstream tasks on resource-limited devices. The authors introduced grouped connections with re-parameterization technology into the training process. The experimental results on serval vision tasks demonstrate the effectiveness of the proposed method.


**Summary Of The Review:**

As written in the Strength And Weaknesses, the strictly restricted application scenario lowers the practical value of the proposed method.

---

> ### Author Response · Authors · 2022-11-17
> **Response for Reviewer HhUN**
>
> Thanks very much for your questions and valuable suggestions.
>
> **Question 1**: Strictly restricted application scenario. The author introduces the application scenario in the abstract where the resource-limited devices cannot store a full copy of parameters but could provide computation to train/fine-tune the large models. It is a bit strange the devices with the ability to train DL models do not have sufficient storage.
>
> **Answer**: Thank you for the question!  As we have stated in the introduction section, we would like to emphasize the application scenario of our consolidator from two aspects: **effective** adaptation and **efficient** adaptation, which are both important.
> 1.  **Effective**. Recently, the scale of vision models grows faster and faster. To sufficiently leverage the capacity of such huge model, a lot of training data are required, e.g. millions of or even more images. However,    the downstream tasks in real-world usually do not contain such many data. Classic fine-tuning which tunes all parameters in a huge model easily falls into an overfitting situation, leading to inferior performance. Consolidator provides a powerful solution to take full advantage of the rich knowledge contained in the huge models. For example, as in Tab. 1, on VTAB benchmark, fine-tuning merely reaches 68.97% accuracy on average, while our consolidator can reach **76.53%** accuracy on average, **7.56%** higher than that of fine-tuning. We have addressed the fantastic performance of consolidator on effective adaptation and made it clearer in the Abstract and Introduction sections in our revised paper.
> 2. **Efficient**. We agree with you that the storage space is usually sufficient on the devices with the ability to **train** DL models. However, in application scenarios, the huge DL models are usually trained/fine-tuned on a computation center, e.g. a powerful server with 8 NVIDIA V100 GPUs and sufficient disk space, and then deployed and doing inference on a resource-limited embedded device, e.g. a typical embedded AI device Jetson TX1[b] with merely 16GB storage, 1 GPU and 4GB GPU memory. We focus on the deployment phase. A ViT-L contains 307M 32-bit floating point parameters, and thus occupies 307 M $\times$ 32 bit $\div$ 8 bit/Byte $\div$ 1024 MB/GB $\approx$ 1.2GB storage space. A larger ViT-H occupies about 2.47GB. Clearly, the storage space of such embedded devices is not sufficient especially considering that there are plenty of tasks to be adapted.
>
> **Question 2**: The proposed method is irrelevant to Transformer. It is closer to the re-parameterization technology.
>
> **Answer**: Thanks for the suggestion. We agree with you that consolidator is not restricted for transformers and can be applied to various vision models as well, e.g. CNNs and MLPs. According to the comment of Reviewer ULMu, we have supplied such additional experiment in Tab. 4 in our revised paper. For re-parameterization technology, we offer a comprehensive discussion in the "inference-efficient structures" part of the Related Work section. To better identify the application scenario of our consolidator, we change the title of our paper from "Consolidator: Mergable Adapter with Grouped Connections for Vision Transformer" to "Mergable Adapter with Grouped Connections for Visual Adaptation", and change the corresponding descriptions in our revised paper.
>
> **Question 3**: Some recent literature[a] on the adaptation of Transformers is missing.
>
> **Answer**: Thanks for the supplement. In our revised paper, we have discussed two latest works published at NIPS'22, AdaptFormer[a] and SSF[c], in the Related Work section and added the experimental results of [a][c] in Tab. 1, Tab. 2 and Tab. 3 for further comparisons. Our result still outperforms their results by a clear margin. Below are the main additional results.
>
> Supervised ViT-B:
> | Method | VTAB-1k | Full data |
> | - | - | - |
> | AdaptFormer | 74.82 | 88.76 |
> | NOAH | 75.48 | - |
> | SSF | 75.69 | 88.60 |
> | Consolidator | **76.53** | **89.31** |
>
> MoCo v3 ViT-B:
> | Method | VTAB-1k | Full data |
> | - | - | - |
> | AdaptFormer | 74.03 | 86.26 |
> | NOAH | 73.55 | - |
> | SSF | 51.41 | 80.73 |
> | Consolidator | **74.71** | **86.41** |
>
> [a] AdaptFormer: Adapting Vision Transformers for Scalable Visual Recognition, NeurIPS 2022.
>
> [b] https://developer.nvidia.com/embedded/jetson-tx1
>
> [c] Scaling & Shifting Your Features: A New Baseline for Efficient Model Tuning, in NeurIPS2022.

---

> ### Author Response · Authors · 2022-12-11
> **Dear Reviewer HhUN**
>
> Thank you again for your time spent on this paper. We wonder if your concerns have been solved, and it would be great if you give us further feedback based on our revised version of the manuscript.

---

### Author Response · Authors · 2022-11-17
**A brief description of the paper revision**

Thanks very much for all the reviewers and chairs to provide helpful and insightful comments. After carefully thinking about the suggestions and concerns raised in the comments, we revise our paper accordingly.

The major changes include:

- According to the comments of Reviewer HhUN and Reviewer ULMu, our method is not strictly restricted for vision transformers, and we have already added experiments on MLP and CNN architectures. To better conclude our method, we change the title of our paper from "Consolidator: Mergable Adapter with Grouped Connections for Vision Transformer" to "Mergable Adapter with Grouped Connections for Visual Adaptation", and change the corresponding descriptions in the main text as well.
- According to the comments of Reviewer HhUN and Reviewer zLNL, we slightly revise the Abstract and Introduction sections to convey the application scenario of our method more accurately. The key change is to further emphasize that consolidator tuning can not only reach parameter- and inference-efficient visual adaptation (outperforms full fine-tuning within 0.35% parameters and without extra inference cost), but can also reach significantly effective visual adaptation (outperforms full fine-tuning by 7.56 accuracy on VTAB-1k benchmark).
- According to the comments of Reviewer ULMu, we add the results of other tuning methods in Tab. 4. And we also add the experimental results of AS-MLP and ConvNeXt, to further demonstrate the effectiveness of our method on different types of architectures other than vision transformers.
- According to the comments of Reviewer HhUN, Reviewer ULMu and Reviewer 2eph, we add two most recent state-of-the-art tuning methods AdaptFormer and SSF that are both published at NIPS'22 in Tab. 1. For a fair comparison on VTAB-1k benchmark, we mostly follow the training configuration of SSF to run our experiments and update the newest results in our revised paper. Moreover, we provide the additional results of AdaptFormer and SSF on full data setting in Tab. 2, and provide the additional results for a self-supervised model MoCo v3 in Tab. 3 as well.
- According to the comments of Reviewer ULMu and Reviewer zLNL, we add a figure in the appendix to show the training efficiency of our method. As shown in Fig. 3, we compare the training speed of the relevant tuning methods across various storage budgets.
- We fix some typos and grammatical issues.

If you have any other questions or suggestions, please inform us.

---

### Decision · Program_Chairs · 2023-01-20

**Decision:**

Accept: poster

**Justification For Why Not Higher Score:**

Combing several designed modules makes a system-like algorithm, which is suitable for a poster.

**Justification For Why Not Lower Score:**

The raised issues are addressed in the technical part and experimental parts. The proposed method is somewhat novel and new.

**Metareview: Summary, Strengths And Weaknesses:**

This submission received mixed reviews. The raised issues include restricted applications, limited novelty, generalizations to other encoder backbones, and insufficient comparisons to other adapters while also a lack of large model evaluations. During the rebuttal phase, the authors addressed all the raised issues one-by-one. After carefully checking all the reviews and rebuttals, the AC feels that most of the raised issues, including novelty discussion upon prior arts, more experimental validations, and technical presentations, are significantly addressed. From the methodology itself, this is not a straightforward design of parameterization. The grouped connections, channel reorder, and other paradigms are carefully proposed to gradually improve the reparameterization process. Considering the well-addressed issues and the technical contribution, the AC recommends acceptance. The authors shall recheck all the raised issues and make sure them to be solved in the final version.

**Note From Pc:**

if the above contains the word "oral" or "spotlight" please see: "oral" presentation means -> notable-top-5% and "spotlight" means -> notable-top-25%. As stated in our emails, we are disassociating presentation type from AC recommendations